# Giardiasis Alters Intestinal Fatty Acid Binding Protein (I-FABP) and Plasma Cytokines Levels in Children in Brazil

**DOI:** 10.3390/pathogens9010007

**Published:** 2019-12-19

**Authors:** Tiara Cascais-Figueiredo, Phelipe Austriaco-Teixeira, Maria Fantinatti, Maria Luciana Silva-Freitas, Joanna Reis Santos-Oliveira, Camila H. Coelho, Steven M. Singer, Alda Maria Da-Cruz

**Affiliations:** 1Laboratório Interdisciplinar de Pesquisas Médicas, Instituto Oswaldo Cruz, Fundação Oswaldo Cruz (FIOCRUZ), Rio de Janeiro 21040-900, Brazil; tiara_ted@hotmail.com (T.C.-F.); phelipe.teixeira@ioc.fiocruz.br (P.A.-T.); fantinatti@ioc.fiocruz.br (M.F.); maria.freitas@ioc.fiocruz.br (M.L.S.-F.); joanna.oliveira@ifrj.edu.br (J.R.S.-O.); 2Núcleo de Ciências Biomédicas Aplicadas, Instituto Federal de Educação, Ciência e Tecnologia (IFRJ), Rio de Janeiro 20061-002, Brazil; 3National Institute of Science and Technology on Neuroimmunomodulation (INCT-NIM), Conselho Nacional de Desenvolvimento Científico e Tecnológico (CNPq), Brasilia 71605-001, Brazil; 4Laboratory of Malaria Immunology and Vaccinology, National Institutes of Allergy and Infectious Diseases, National Institutes of Health, Bethesda, MD 20892, USA; camila.coelho@nih.gov; 5Biology Department, Georgetown University, Washington, DC 20057, USA; steven.singer@georgetown.edu; 6Disciplina de Parasitologia-DMIP, Faculdade de Ciências Médicas, Universidade do Estado do Rio de Janeiro (UERJ), Rio de Janeiro 20550-170, Brazil

**Keywords:** *Giardia lamblia*, giardiasis, children, I-FABP, cytokines, IL-17, *Giardia* assemblages

## Abstract

Giardiasis is an intestinal infection caused by ingestion of water or food contaminated with cysts of *Giardia lamblia*. Susceptibility is higher in children and overall prevalence can reach up to 90% in low-income areas, although outbreaks are also reported in developed countries. Both parasite and immune-mediated epithelial damage has been observed in vitro and in animal models. However, whether enterocytes are directly damaged during infection is not entirely known. Our goal was to identify whether plasma levels of intestinal fatty acid binding protein (I-FABP), a marker of enterocyte damage, are related to the immune response in giardiasis. Blood plasma was collected from 31 children (19 *Giardia*-positive) from a public day care in Rio de Janeiro, Brazil. The levels of I-FABP were increased in *Giardia*-infected children compared to children without detectable infection. There was no difference in I-FABP levels in giardiasis caused by different genetic assemblages of *Giardia*. Levels of IL-8 were decreased, while there was a trend to elevated IL-17 in the *Giardia*-positive children. A positive correlation was observed between I-FABP and IL-17 levels as well as TNF, suggesting that epithelial damage can be related to cytokine production during giardiasis. These results help elucidate the relationship between the disruption of the intestinal mucosal barrier and immune responses to *G. lamblia* in children.

## 1. Introduction

Giardiasis is a worldwide intestinal infectious disease caused by transmission of water or food contaminated with cysts of *Giardia lamblia,* and exhibits both anthroponotic and zoonotic transmission. The real incidence of giardiasis is likely underestimated, since asymptomatic and even symptomatic patients frequently do not seek care. In addition, the low sensitivity of microscopic detection methods, commonly utilized in middle and low-income areas, further contributes to underreporting. Global initiatives examining the etiology of diarrhea identified *Giardia* as the second most common pathogen detected among children 12–24 months old [1] and one of the top four contributors to stunting, globally [2]. In Brazil, we showed that the prevalence of *Giardia* can reach 78% [3].

Eight assemblages of *G. lamblia* (A to H) have been described based on genetic differences. Assemblages A and B are the most commonly found in humans, although both also infect a variety of animals. Our group has previously identified assemblage E in humans [4], but there is no conclusive evidence of a correlation between clinical outcome or parasite burden and assemblage. 

*Giardia* trophozoites colonize the luminal surface of the small intestine and attach to intestinal epithelial cells (IEC). Disease is thought to occur due to damage to the epithelial barrier, mediated by both the parasite and the immune response. The disruption of tight junctions, shortening of microvilli, epithelial permeability and altered intestinal motility have all been observed [5]. Several studies have analyzed cytokines in murine giardiasis, and a consensus has formed on the importance of IL-17 for protective immunity [6]. In adult human patients who recovered from giardiasis, IL-17 was upregulated when effector memory CD4^+^ T cells were restimulated in vitro with *Giardia* antigens [7]. However, roles for other cytokines in pathogenesis of giardiasis are much less clear, and few studies have addressed this issue in humans [8,9]. 

*Giardia*-secreted metabolites [10] along with cytokines and cytotoxic T cell responses are related to epithelial damage and the apoptosis of enterocytes [11]. However, whether enterocytes are directly damaged is not known. Intestinal fatty acid binding protein (I-FABP) is an intracellular protein that acts on the metabolism of long chain fatty acids and is abundantly expressed in the cytosol of epithelial cells in the small intestinal mucosa. Increased plasma levels of I-FABP have been associated with intestine-specific injury in several diseases [12]; however, studies of I-FABP in giardiasis are lacking. 

Our goal was to identify whether enterocyte damage is connected to alterations in immune responses in children living in a low-income setting. Here, we show that plasma I-FABP levels are increased in *Giardia*-infected children and that there is a correlation between I-FABP levels and systemic cytokine levels. However, we did not observe any relationship with a specific *Giardia* genetic assemblage.

## 2. Results

### 2.1. Clinical and Hematological Profiles of Giardia Lamblia-Infected Children

In an observational study performed in a government-run day care in Rio de Janeiro city, we identified 19 children infected with *G. lamblia* and 12 without laboratory evidence of intestinal parasites. The assemblages of *G. lamblia* identified were A (*n* = 11), B (*n* = 6) and E (*n* = 2). 

Alteration in normal stool consistency was a frequent occurrence among day care children. Thus, no association between diarrhea and infection by *G*. *lamblia* could be established. The children’s anthropometric statuses, evaluated according to the WHO Z scores (length for age, weight for age and weight for length), showed that 28 out of 33 children were eutrophic, but only two of them were underweight. 

The mean levels of hemoglobin in *Giardia*-positive preschoolers (11.50 g/dL (11.10–12.20 g/dL)) were similar to *Giardia*-negative (11.90 g/dL (11.28–12.53 g/dL)). Only one *Giardia*-negative child presented with moderate anemia. Similar results were observed for the other hematological parameters related to anemia. To investigate a possible influence of *G. lamblia* on immune cells we assessed leukocyte profiles. No differences were observed for lymphocyte, neutrophil or monocyte counts when *Giardia*–positive and *Giardia*–negative groups were compared. Seven out of 19 *Giardia*-positive children presented mildly increased eosinophil counts, and four out of 12 *Giardia*-negative children showed mild or moderate increases (*p* = 0.80) (Table 1). 

### 2.2. Plasma Cytokine Profiles of Giardia Lamblia-Infected Preschoolers Differ from Those Not Infected by Giardia

Plasma levels of 15 cytokines or chemokines were quantified (Appendix A). Compared to *Giardia*-negative (*n* = 8), the *Giardia*-positive group (*n* = 17) showed significantly lower levels of IL-8 (*p* = 0.003). The *Giardia*-positive group also showed a trend to higher levels of IL-17 (*p* = 0.09) and IL-10 (*p* = 0.08) and the IL-17 levels were significantly different (*p* = 0.047) when evaluated using a Student’s *t*-test. No difference in IFN-γ levels was observed between the two groups, but the IFN-γ/IL-10 ratios were also significantly lower (*p* < 0.05) in the *Giardia*-positive group in comparison to *Giardia*-negative (Figure 1). 

### 2.3. Intestinal Fatty AcidBinding Protein (I-FABP) Levels Are Increased in Giardia Lamblia-Infected Children

Considering that enterocyte damage has been implicated in increased intestinal permeability and microbial translocation into the circulation [13] we evaluated whether *G. lamblia* infection could increase I-FABP levels, a known biomarker for damaged intestinal epithelial cells. *Giardia*-positive preschoolers presented significantly higher I-FABP levels (1274 pg/mL (622–2004 pg/mL); *n =* 17) than *Giardia*-negative children (741 pg/mL (422–960 pg/mL); *n =* 12; *p* = 0.05) (Figure 2).

As both low and high I-FABP levels were observed among *Giardia*-positive subjects, we decided to evaluate a possible relationship between enterocyte damage and the *G. lamblia* assemblages. There was no difference in I-FABP levels comparing children infected by assemblage A (1591 pg/mL (615–1882 pg/mL); *n =* 11) or assemblage B and E (1032 pg/mL (637–2431 pg/mL); *n =* 6) (Figure 2). We next asked whether the epithelial damage could be related to levels of systemic cytokines. A positive correlation was observed between I-FABP and IL-17 levels (r = 0.453; *p* < 0.03) as well as TNF (r = 0.491; *p* < 0.02) (Figure 2). Four *Giardia*-infected children showed I-FABP levels higher than 2000 pg/mL, although no clinical features could be associated with these high levels of I-FABP.

Considering the high variation observed among all the parameters analyzed, we investigated which factors were influencing the elevated I-FABP levels by using a multivariate linear regression analysis. The model showed a trend that *Giardia* infection influenced I-FABP levels (*p* = 0.071) (Table 2), supporting an association between enterocyte damage and giardiasis. This correlation was independent of the levels of cytokines.

## 3. Discussion

The major findings of this study are that children infected with *Giardia* exhibited elevated plasma levels of IL-17 and reduced levels of IL-8 compared with children not infected with intestinal parasites. Infected children also had lower IFN-γ/IL-10 ratios and elevated levels of I-FABP that correlated with both IL-17 and TNF. The increase in I-FABP levels, however, was not related to the specific assemblage of *Giardia* with which they were infected.

Elevated in vitro IL-17 responses have been reported in a cohort of adult travelers returning to Denmark with giardiasis [7] and IL-17 has been shown to be essential for proper control of *G. muris* and *G. lamblia* infections in mice [6]. This is the first report of which we are aware showing IL-17 production in children with giardiasis.

Interleukin-8 is a chemokine that recruits neutrophils to sites of infection and inflammation. Recent work has indicated that proteases secreted by *Giardia* are able to degrade IL-8 in vitro [14]. Moreover, granulocyte responses after intracolonic administration of *C. difficile* toxin were attenuated in mice infected with *G. lamblia* strain NF [15]. This is the first report, to our knowledge, of reduced systemic levels of IL-8 in giardiasis. It would be interesting to determine if mucosal cytokine levels are also diminished.

Interferon-γ is a cytokine often associated with immunopathology of giardiasis. In a study of patients in Iran, symptomatic patients had higher levels of IFN-γ than uninfected controls, while patients with asymptomatic infections did not [8]. We were unable to classify the *Giardia*-infected children in this study as being symptomatic or asymptomatic. Nevertheless, we found a reduced ratio of IFN-γ/IL-10 in the infected children. This is consistent with the lack of inflammation often reported in giardiasis.

While giardiasis has previously been shown to impact epithelial permeability [5], specifically, lactulose:mannitol test ratios, other markers of intestinal damage, have not been widely addressed in this infection. Our results are the first to indicate that children with *Giardia* infections have elevated levels of I-FABP, a marker for epithelial cell damage. Moreover, although I-FABP levels were independent of the genetic assemblage of *Giardia* present, they did correlate with plasma levels of both IL-17 and TNF. This correlation could indicate that immune responses themselves contribute to the epithelial damage observed or could reflect that epithelial damage facilitates the induction of the Th17 developmental program resulting in both IL-17 and TNF production. Unfortunately, our data were unable to determine a causal relationship between these cytokines and I-FABP levels.

Our study suffers from several limitations. First, limited access to additional biological samples prevents drawing extensive conclusions about the patterns observed and a larger cohort should be used to confirm these observations. We did not perform systematic surveillance on diarrhea status, although alteration in stool consistency was frequently noted among the children studied. While we were able to exclude concurrent parasitic infections and only *Giardia*-infected children were enrolled, we did not evaluate whether bacterial or viral infections were present that could also contribute to systemic levels of cytokines and/or I-FABP. Finally, because of the cross-sectional study design we were unable to determine whether the children enrolled in the study had been previously infected with *Giardia*, since a secondary immune response could lead to much greater cytokine production.

In summary, our data provides novel information on levels of cytokines and I-FABP in children with giardiasis and suggests a link between mucosal damage and the immune response. Further studies to identify the mechanisms underlying this correlation are needed.

## 4. Material and Methods

### 4.1. Study Design and Participants and Ethical Aspects

The cross-sectional study was conducted in a community (urban slum) of Rio de Janeiro, Brazil in 2015, from February to December. The area has a limited drinking water supply, no sewage network coverage and few streets are paved. Stray animals, such as dogs, cats, rodents, pigs, horses and cattle are commonly found moving throughout the location. The majority of inhabitants are from the low socioeconomic strata. 

For the selection of preschoolers infected by *G. lamblia* (*Giardia*-positive) or not (*Giardia*-negative), stool samples were obtained from 105 children (ages varying from 10 months to four years) and used in parasitological and PCR assays for the detection of *Giardia* DNA). A collection device accompanied by instructions was provided for the guardians of children and nursery volunteers. One stool sample from each participant was examined for intestinal protozoa and helminths by three methods: Ritchie, Kato–Katz and spontaneous sedimentation. Children infected or co-infected by other pathogenic intestinal protozoa or helminths were excluded. Thirty-one children (19 *Giardia*- positive and 12 *Giardia*-negative) were selected for the study. The subjects were evaluated for anthropometric parameters. Standard deviation scores (Z-scores) of weight-for-height (WHZ), lenght-for-age (LAZ) and weight-for-age (WAZ) were calculated according to the World Health Organization´s 1978 growth chart. Blood was submitted to hematological exams. Plasma samples were separated, and aliquots were stored at −70°C for immunological analysis.

All procedures were approved by the ethics committee for human research (Instituto Oswaldo Cruz/FIOCRUZ, Brazil, CAAE: 19705613.9.0000.5248). Biological samples were collected after informed consent was obtained from the guardians of the children.

### 4.2. Molecular Detection and Characterization of Giardia Lamblia

The stool DNA extraction was conducted using the QIAmp DNA Stool Mini Kit (Qiagen GmbH., Germany), with modifications [3]. Conserved gene fragments from glutamate dehydrogenase (*gdh*) and beta-giardin (*βgia*) were amplified by polymerase chain reaction (PCR) using the primers previously reported [3]. The amplicons obtained for each pair of primers were purified using the NucleoSpin® Gel and PCR Clean-up kit (Macherey-Nagel GmbH and Co. KG, Germany). The purified products were sequenced using the ABI Prism™ BigDye Terminator Cycle Sequencing kit. Electropherograms were analyzed using Chromas 2.4 (Technelysium Pty Ltd., South Brisbane, Australia). Characterization of the sequences was performed using the Basic Local Alignment Search Tool for nucleotides (BLASTn), and the contigs were obtained by the CAP3 Sequence Assembly Program. Nucleotide sequences of *gdh* and *βgia* were aligned by the CLUSTAL W algorithm from Molecular Evolutionary Genetics Analysis (MEGA) output 7.0. The phylogenetic analysis was performed on the MEGA and the range estimation equations used were JIN and NEI (Kimura 2-parameter model). Phylogenetic trees were constructed using the neighbor-joining algorithm, with bootstrap analysis (1000 replicates). The sequences from the new isolates were aligned using reference sequences of *G. lamblia* from GenBank. The sequences obtained were deposited in GenBank: genotype A (*gdh*: MN541586, MN541659, MN541674, MN5416956, MN541648, MN541661, MN541666, MN541658, MN541657 and MN541804; *βgia*: MN541703, MN541775, MN541806, MN541790, MN541772, MN541764, MN541777, MN541782, MN541774, MN541773 and MN541804); genotype B (*gdh*: MN541687, MN541686, MN541643, MN541684, MN541678 and MN541668; *βgia*: MN541803, MN541802, MN541759, MN541800, MN541794 and MN541784), genotype E (*gdh:* MN541650 and MN541637; *βgia*: MN541766 and MN541754)

### 4.3. Quantification of Intestinal Fatty Acid Binding Protein (I-FABP) and Cytokines in Plasma

I-FABP levels were determined by ELISA (Duo Set; R&D Systems, Minneapolis, MN, USA). The results were expressed as pg/mL, and the minimum detection limit was 31.2 pg/mL.

For cytokine measurement, a multiplex biometric immunoassay containing fluorescently dyed microbeads was used (BioRad Laboratories, Hercules, CA, USA). The following cytokines were quantified: IFN-γ, TNF, IL-1β, IL-2, IL-4, IL-5, IL-6, IL-7, IL-8, IL-10, IL-12, IL-13, IL-17, MCP-1 and MIP-1β. Cytokine levels were calculated by Luminex Technology (Bio-Plex Workstation; Bio-Rad Laboratories, Hercules, CA, USA). The analysis of data was performed using software provided by the manufacturer (Bio-Rad Laboratories, Hercules, CA, USA). A range of 0.51–8000 pg/mL of recombinant cytokines was used to establish standard curves and the sensitivity of the assay. The results were expressed as the median fluorescent intensity (MFI) [16]. 

### 4.4. Statistical Analyses

Statistical analyses were performed using GraphPad Prism software (version 7.0, San Diego, CA, USA). Mann–Whitney tests were performed for comparisons between two groups. A Student’s *t*-test was additionally used when Mann–Whitney tests indicated a trend and the data were normally distributed. The Spearman test was used for correlation analysis and Pearson’s test was used for correlation matrix analysis. Continuous variables were expressed as medians and interquartile ranges (IQRs). Differences were considered statistically significant when a *p*-value was equal or below 0.05. Multivariate linear regression analyses (software IBM SPSS, version 22.0, IBM, USA) were used to determine the influence of intervening variables on the levels of I-FABP. *Giardia* infection (positive or negative) and MFI levels of cytokines were considered as independent variables. 

## Figures and Tables

**Figure 1 pathogens-09-00007-f001:**
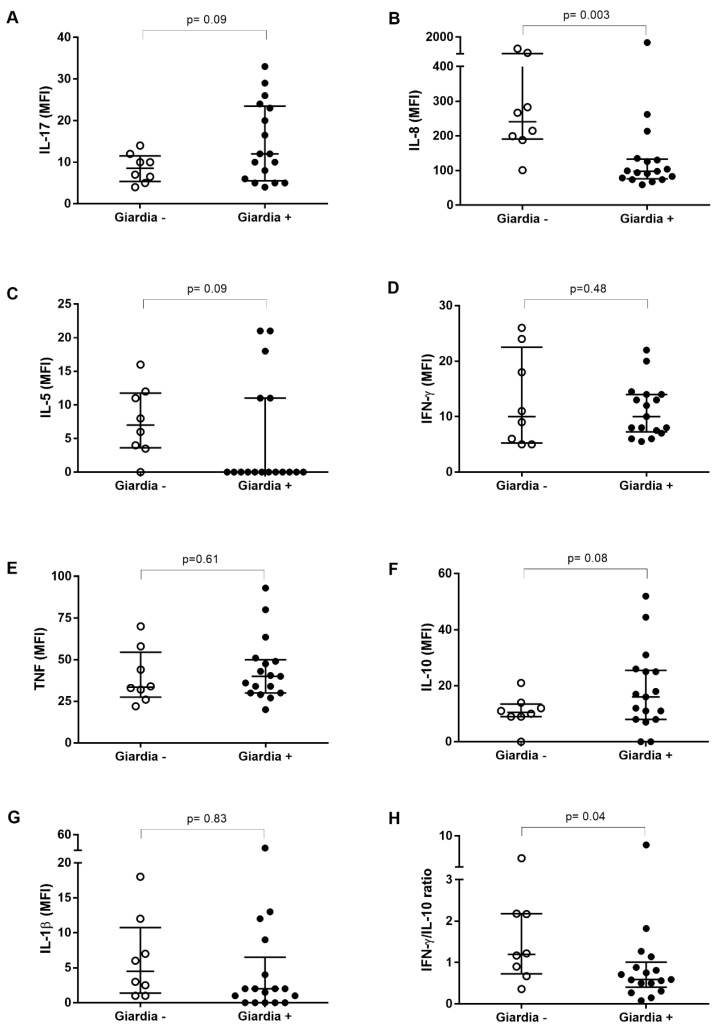
The cytokine profiles of *Giardia lamblia*-infected children from a Brazilian area. Plasma levels of (**A**) IL-17, (**B**) IL-8, (**C**) IL-5, (**D**) IFN-γ, (**E**) TNF, (**F**) IL-10 and (**G**) IL-1β in children with giardiasis were determined using Luminex beads. The IFN-γ/IL-10 ratio (**H**) is also shown. Each point corresponds to one subject. Results are expressed as median fluorescence intensity (MFI). Medians (horizontal bars), quartiles (range 25–75%) and results of Mann–Whitney statistical tests are shown.

**Figure 2 pathogens-09-00007-f002:**
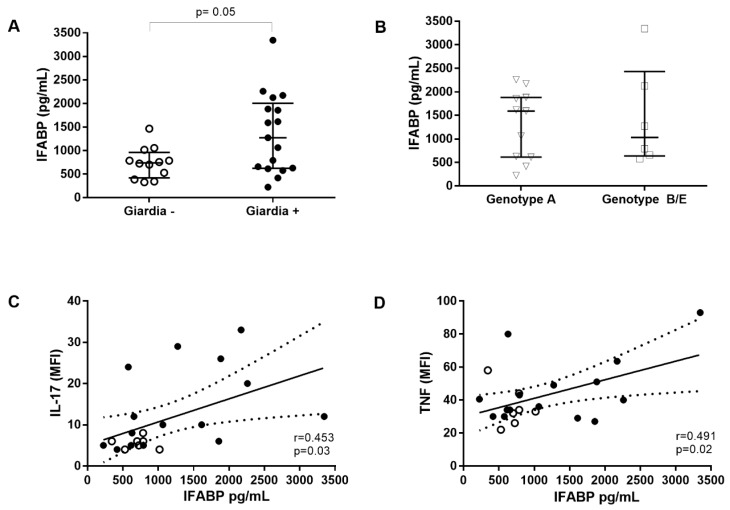
Intestinal fatty acid binding protein (I-FABP) levels and the correlation with systemic cytokine profiles and assemblages in *Giardia*-infected children in a Brazilian area. (**A**) I-FABP levels are increased in *G. lamblia*-infected pre-schools children. (**B**) Different assemblages (A, B or E) of *G. lamblia* are not associated to I-FABP levels. Correlation between I-FABP and IL-17 (**C**) or TNF (**D**) levels. Levels of I-FABP were elevated in *Giardia*-positive subjects (●) in comparison to *Giardia*-negative (○). Each point represents one subject. The horizontal bars indicate the median values and verticals bars indicate interquartile ranges (25–75%).

**Table 1 pathogens-09-00007-t001:** Demographic data and hematological exams of the *Giardia*-positive and *Giardia*-negative children.

Parameters Analysed		Preschoolers	
		*Giardia* Positive	*Giardia* Negative
Age	(Months)Median IQ interval	27.9 [25.5–31.7]	36.2 [29.6–41.2]
Gender	Male	9	2
	Female	10	10
Hemoglobin	Normal	15	10
	Mild (10–10.9 g/dL)	4	1
	Moderate (7–9.9 g/dL)	0	1
	Severe (<7 g/dL)	0	0
Eosinophils	Normal (<500 cells/mm^3^)	11	8
	Mild (501–1500 cells/mm^3^)	7	2
	Moderate (1501–5000 cells/mm^3^)	1	2
	Severe (>5000 cells/mm^3^)	0	0
Neutrophils	Normal (1600–7150 cells/mm^3^)	18	12
	Increased (>7151 cells/mm^3^)	1	0
Monocytes	Normal (80-1100 cells/mm^3^)	15	10
	Increased (>1101 cells/mm^3^)	4	2
Lymphocytes	Normal (880–4950 cells/mm^3^)	14	8
	Increased (>4951 cells/mm^3^)	5	4

**Table 2 pathogens-09-00007-t002:** Multivariate linear regression analysis to evaluate the association between the levels of intestinal fast acid bound protein (I-FABP) and independent variables (cytokines) in *Giardia*-positive preschoolers.

Independent Variables	Dependent Variable I-FABP Levels (pg/mL)		
	Coef ^1^	SE ^2^	P
*Giardia* infection (positive or negative)	0.396	321.88	0.071
IL-17 *	0.227	18.86	0.349
TNF *	0.437	9.80	0.135
IL-8 *	−0.045	0.55	0.833
IL-10 *	−0.341	14.29	0.215

^1^ Coef—Correlation coeficient; ^2^ SE—standard error; * MFI—median fluorescence intensity; IL—interleukin, TNF—tumor necrosis factor, I-FABP—intestinal fast acid bound protein.

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
