# Peer review of "Giardiasis Alters Intestinal Fatty Acid Binding Protein (I-FABP) and Plasma Cytokines Levels in Children in Brazil"

_pathogens, 2019, doi:10.3390/pathogens9010007_

Round 1
Reviewer 1 Report
The protozoan Giardia lambda may cause acute or chronic diarrhea in humans and various animal species. Data from current literature indicate that young individuals are particularly susceptible to this intestinal parasite. In the present study, Cascais-Figueireido et al. investigated Giardia lamblia-infected with non-infected Brazilian preschool children regarding a set of systemic parameters that are relevant for the immunobiology and/or supposedly also the pathogenesis associated with the infection. Most importantly, the authors identified a trend of a decrease of interleukin (IL)-5 and IL-8 production as well as an increase in IL-17 production in infected individuals. Furthermore, these increased IL-17 levels seemed to correlate with a slight (but apparently statistically significant) rise in the level of the fatty acid binding protein (I-FABP), a marker for damage in of enterocytes. Based on these observations, the authors suggested a link between damage of enterocytes and anti-parasitic immunity.
Basically, the data presented in this study are of considerable interest because they support previous observations in experimental murine infection model where IL-17 turned out to be a key regulatory factor in the immunological (and physiological) response against the parasite infection. Furthermore, they are in agreement with recent reports describing the occurence of elevated levels of IL-17 in giardiasis patients. Unfortunately, however, I have serious doubts about the significance of respective data because the statistical analysis applied in the present study seem to me inappropriate for two reasons: 1.) To take into account the multiplicity of dependent parameters (e.g. 8 different cytokines), more appropriate statistical tests should be applied (e.g. PCA, PLS-DA). Moreover, when comparing a single parameter (e.g. interleukin x) between the two groups, the significance level p should be corrected for the multiplicity of parameters. 2.) The variation within one group is high for all dependent parameters. Therefore, comparing the levels of given cytokines within one group via an appropriate analytical design (e.g. randomized block design) would be required. Then, cytokines showing high variability independent of giardial infection could be excluded as potential markers.
In summary, additional statistical analyses e.g. including those listed above will be necessary to verify the conclusions drawn in the present report.
Author Response
RESPONSE TO REVIEWER 1
“ … Based on these observations, the authors suggested a link between damage of enterocytes and anti-parasitic immunity. Basically, the data presented in this study are of considerable interest because they support previous observations in experimental murine infection model where IL-17 turned out to be a key regulatory factor in the immunological (and physiological) response against the parasite infection. Furthermore, they are in agreement with recent reports describing the occurrence of elevated levels of IL-17 in giardiasis patients.…”
Response: We thank the reviewer for recognizing the importance of this contribution.
Unfortunately, however, I have serious doubts about the significance of respective data because the statistical analysis applied in the present study seem to me inappropriate for two reasons: To take into account the multiplicity of dependent parameters (e.g. 8 different cytokines), more appropriate statistical tests should be applied (e.g. PCA, PLS-DA). Moreover, when comparing a single parameter (e.g. interleukin x) between the two groups, the significance level p should be corrected for the multiplicity of parameters. 2.) The variation within one group is high for all dependent parameters. Therefore, comparing the levels of given cytokines within one group via an appropriate analytical design (e.g. randomized block design) would be required. Then, cytokines showing high variability independent of giardial infection could be excluded as potential markers. In summary, additional statistical analyses e.g. including those listed above will be necessary to verify the conclusions drawn in the present report.
Response: We agree with the reviewer. To address this point, we performed a multivariate linear regression analysis using software IBM SPSS to determine influence that of intervenient factors on the levels of I-FABP. Giardia infection (positive or negative) and levels of cytokines were considered as independent variables. The results showed a p value of 0.071 and no statistical significance for cytokines.
Then, the following modifications were done:
a) the statistical analysis was described in the “Material and methods” section (Lines 257-259); b) a table ( Table 2 – lines 141-147) and the following sentence were added to the text (Lines 135-139): “Considering the high variation observed among all the parameters analyzed, we investigated which factors were influencing the elevated I-FABP levels by using a multivariated linear regression analysis. The model showed a trend that Giardia infection influenced I-FABP levels (p = 0.071) (Table 2), supporting an association between enterocyte damage and giardiasis. This correlation was independent of the levels of cytokines.”

Reviewer 2 Report
In the manuscript „Giardiasis alters intestinal fatty acid binding protein (I-FABP) and plasma cytokines levels in children in Brazil” of Tiara Cascais-Figueiredo and colleagues, the authors reported on the finding that Giardia-infected children, from 10 months to four years old, plasma levels of IL-5 25 and IL-8 were decreased, while the key cytokine for parasite control, IL-17, was increased. Furthermore, the marker for damage in intestinal enterocytes the intestinal fatty acid binding protein (I-FABP), was observed to be also higher in Giardia-infected than in uninfected children.
This reviewer feels that the study provides a basic set of information on the presented topic. However, the manuscript is well written and due to lack of knowledge the information would be helpful for Giardia diagnostics. It is important to modify the manuscript in order to state that the observations that were made are not backed on a statistical set of data.
Abstract: Please rephrase and extend the information provided in the abstract. Currently the abstract presents only some partial information but the story and the background is lacking.
Lines 66-68: Please provide information to the time period in which the cases were analyzed.
Please use a unique spelling for Giardia(-)positive/negative
Please also include a short section to the limited availability of the samples and that the results are not based on a reliable set of cases. Due to the limited set of cases analyzed the study provides only observations!
Provide the Accession numbers of the sequences in the manuscript.
Please revise spelling and typos carfully.
Author Response
RESPONSE TO REVIEWER 2
“ … However, the manuscript is well written and due to lack of knowledge the information would be helpful for Giardia It is important to modify the manuscript in order to state that the observations that were made are not backed on a statistical set of data.
Response: We agree with the reviewer. This point was also questioned by the Reviewer 1. To address this point, we performed a multivariate linear regression analysis using software IBM SPSS to determine influence that of intervenient factors on the levels of I-FABP. Giardia infection (positive or negative) and levels of cytokines were considered as independent variables. The results showed a p value of 0.071 and no statistical significance for cytokines.
Then, the following modifications were done:
a) the statistical analysis was described in the “Material and methods” section (Lines 257-259); b) a table ( Table 2 – lines 141-147) and the following sentence were added to the text (Lines 135-139): “Considering the high variation observed among all the parameters analyzed, we investigated which factors were influencing the elevated I-FABP levels by using a multivariated linear regression analysis. The model showed a trend that Giardia infection influenced I-FABP levels (p = 0.071) (Table 2), supporting an association between enterocyte damage and giardiasis. This correlation was independent of the levels of cytokines.”
Abstract: Please rephrase and extend the information provided in the abstract. Currently the abstract presents only some partial information but the story and the background is lacking.
Response: We have extended the content including more information on the background and also detailing the results.
Please provide information to the time period in which the cases were analyzed.
Response: The time period was added as requested “The cross-sectional study was developed in a community (urban slum) of Rio de Janeiro, Brazil in 2015, from February to December” (Line 195)
Please use a unique spelling for Giardia(-)positive/negative
Response: We have modified all the spellings for this case to Giardia-positive or Giardia-negative.
Please also include a short section to the limited availability of the samples and that the results are not based on a reliable set of cases. Due to the limited set of cases analyzed the study provides only observations!
Response: We agree with the reviewer. We have modified (see highlighted text) the following sentence of the manuscript text (Lines179-181): “First, limited access to additional biological samples, prevents drawing extensive conclusions about the patterns observed and a larger cohort could be used to confirm these observations.”
Provide the Accession numbers of the sequences in the manuscript.
Response: We included this information (Lines 231-237).
Please revise spelling and typos carefully.
Response: We appreciate this comment and have submitted the new version to English grammar and spelling revisions. All the revisions are highlighted on the text.

Round 2
Reviewer 1 Report
In the revised version of the manuscript, the authors adequately addressed my major critisism regarding the statistical analyses applied in the study
Please correct in Table 2 (line 142) "Girdia"
Author Response
Response to Reviewer 1
Thanks for your comments. The modification on the table title was done as requested
Reviewer 2 Report
The majority of the requested revision were addressed.
Author Response
Thanks for your reply